# Influence of Calcination and Cation Exchange (APTES) of Bentonite-Modified Reinforced Basalt/Epoxy Multiscale Composites’ Mechanical and Wear Performance: A Comparative Study

**DOI:** 10.3390/ma17194760

**Published:** 2024-09-27

**Authors:** Saurabh Khandelwal, Vivek Dhand, Jaehoon Bae, Taeho Kim, Sanghoon Kim

**Affiliations:** 1Department of Mechanical Engineering, College of Engineering, Kyung Hee University, Yongin 17104, Gyeonggi-do, Republic of Korea; saurabhkh18@gmail.com; 2Department of Mechanical Design Engineering, Chonnam National University, Yeosu-si 59626, Jeollanam-do, Republic of Korea; vivekdhand2012@gmail.com; 3Department of Architectural Design, Chonnam National University, 50 Daehak-ro, Yeosu-si 59626, Jeollanam-do, Republic of Korea; skycity-bjh@jnu.ac.kr; 4Department of Marine Production Management, Chonnam National University, 50 Daehak-ro, Yeosu-si 59626, Jeollanam-do, Republic of Korea; 5Smart Aquaculture Research Center, Chonnam National University, 50 Daehak-ro, Yeosu-si 59626, Jeollanam-do, Republic of Korea

**Keywords:** bentonite, swell behavior, surface area, basalt fiber-reinforced epoxy composite

## Abstract

In this study, bentonite clay was modified through silane treatment and calcination to enhance its compatibility with basalt fiber (BF) and epoxy in multiscale composites. The as-received bentonite (ARB) was subjected to silane treatment using APTES, producing silane-modified bentonite (STB), while calcination yielded calcined bentonite (CB). The modified clays were incorporated into basalt fiber-reinforced epoxy (BFRP) composites, which were fabricated using the vacuum-assisted resin transfer method (VARTM). Analytical techniques, including X-ray diffraction (XRD) and Fourier-transform infrared (FTIR) spectroscopy, confirmed the structural changes in the clays. BET surface area analysis revealed a 314% increase in the surface area of STB and a 176% increase for CB. The modified clays also demonstrated reduced hydrophilicity and swelling behavior. Thermogravimetric analysis (TGA) indicated a minimal improvement in thermal stability, with the degradation onset temperatures increasing by less than 3 °C. However, tensile tests showed significant gains, with CB- and STB-reinforced composites achieving 48% and 21% higher tensile strength than ARB-reinforced composites. Tribological tests revealed substantial reductions in wear, with CB- and STB-reinforced composites showing 90% and 84% decreases in the wear volume, respectively. These findings highlight the potential of modified bentonite clays to improve the mechanical and wear properties of basalt fiber–epoxy composites.

## 1. Introduction

Advanced composite materials composed of macroscale fibers and nanomaterials possess greater thermal stability and higher mechanical strength than those of conventional polymer composites. In recent years, nanocomposites have attracted significant research attention because their thermal and mechanical properties are customized to cater to the requirements of different applications, such as automobiles, and their processing costs are lower than those of metal composites. The availability of diverse woven and chopped fibers has facilitated the application of polymer nanocomposites in the aviation and marine domains. Basalt fiber (BF) has become highly valuable in commercial applications because of its low cost, eco-friendly and non-toxic behavior, and superior mechanical and thermal properties compared to those of glass fiber (GF). Lopresto et al. [1] compared the mechanical properties of epoxy composites composed of BF or GF, and their results indicated that the fiber–matrix interface in the BF–epoxy composites was superior. Colombo et al. [2] fabricated composites of BF with vinyl ester and epoxy matrixes separately. The tensile and compressive strengths of the epoxy composites were higher than those of the vinyl ester composites. BF offers several advantages over other fibers, but a few studies have demonstrated the vulnerability of the BF–polymer interface to saline solutions [3]. The addition of microfillers and nanofillers, such as graphene, montmorillonite, and fly ash, to FRP alters the interfacial interactions and the composites’ bulk properties owing to nanostructuring within the matrix [4]. Ji et al. investigated the wear rates of epoxy composites whose surfaces were modified with SiC nanoparticles. Their results indicated significant reductions in the friction coefficient and wear rate of epoxy owing to strong interfacial bonding between the grafted SiC nanoparticles and the epoxy matrix [5].

In previous studies, the mechanical properties of nanocomposites were enhanced using clays, but preprocessing was needed to counteract the poor dispersion and aggregation of these clays in the polymer matrix [6,7]. Owing to their wide availability and high aspect ratio, smectite clay minerals such as montmorillonite (MMT) are widely used as fillers in polymer nanocomposites. Garima et al. investigated the mechanical properties of silane-modified MMT GF–vinyl ester composites [7]. Sung et al. investigated the effects of the MMT volume fraction on the fracture behavior of MMT epoxy nanocomposites [8]. These studies highlighted improvements in the mechanical properties of polymer nanocomposites owing to the hydrophobic behaviour of silane-modified MMT. Silane-modified MMT exhibited improved dispersion in polymer matrixes and contained fewer voids; therefore, it enhanced the mechanical properties of the polymer nanocomposites. However, in these studies, the shrink–swell behavior and specific surface areas (SSA) of the clays, which determine the interfacial interactions and mechanical performance of nanocomposites, were not characterized. In addition, the thermal, mechanical, and wear performance of BF/bentonite/epoxy multiscale composites has rarely been studied. Moreover, even though bentonite clay is a smectite mineral similar to MMT, its shrink index and SSA differ significantly from those of MMT.

Thus, in this study, the SSA, swell index, and interlayer spacing of bentonite clay are altered through interlayer cation exchange and calcination. The effects of these modified bentonite clays on the interfacial adhesion, thermal stability, and tensile strength of multiscale BF/bentonite/epoxy composites are investigated. The friction coefficient and wear track of the composites are further analyzed using a surface profilometer.

## 2. Material and Methods

### 2.1. Materials

Woven basalt fiber (BF) with a mass per unit area of 200 g per square meter, an 11-micrometer diameter, and a thickness of 0.22 mm was acquired from GBF, Jinhua, Zhejiang in China. The epoxy resin and hardener were sourced from Kukdo Chemicals in Seoul, Republic of Korea, specifically labeled EPOKUKDO YD115 for the resin and DOMIDEG(A0533) for the hardener. LOCTITE^®^ FREKOTE 44-NC, provided by Henkel, Rocky Hill, CN, USA, served as the mold release agent. Additionally, bentonite and various other chemicals were procured from Sigma-Aldrich (St. Louis, MO, USA).

### 2.2. Silane Modification and Calcination of Bentonite

First, 50 g of the raw bentonite (ARB) clay was dispersed in 200 mL of ethanol and stirred vigorously for 1 h to prevent particle clumping. Simultaneously, 50 g APTES was stirred in a separate flask with 100 mL of ethanol for 1 h. Afterward, the clay suspension was combined with the APTES solution and stirred continuously for 12 h at 25 °C, maintaining a 1:1 clay-to-APTES ratio. The resulting silane-modified bentonite (STB) clay was washed with ethanol, filtered, and dried for 48 h at 100 °C. Meanwhile, the ARB clay underwent calcination in a programmable furnace, where the temperature was gradually increased from ambient to 800 °C at a rate of 13 °C per minute, followed by a hold time of 61.5 min. After calcination, the clay was finely ground and passed through a 75 µm mesh. Both the silane-treated and calcined clays (STB and CB) were then subjected to further characterization.

### 2.3. Fabrication of Multiscale Basalt Epoxy Composite

The VARTM process was used to create multiscale composites composed of basalt fiber (BF), bentonite clay, and epoxy. The epoxy resin and hardener were mixed in a ratio of 100:55. Bentonite clay (1% by weight) was added to the epoxy and mixed for 30 min, and then the mixture was degassed. The hardener was then added, and the resulting mixture was transferred over seven layers of BF (140 *×* 140 mm^2^) attached to a VARTM mold using vacuum suction. The mold was cured at 86 °C for two hours. After cooling for eight hours, the finished composites were removed from the mold.

### 2.4. Analytical Techniques

X-ray diffraction (XRD) was used to determine the interlayer spacing of all powdered clays (ARB, CB, and STB) in the range of 3–25 degrees at a rate of 2 degrees per minute. Fourier-transform infrared (FTIR) spectroscopy was used to identify the surface functional groups of the clays. The Brunauer–Emmett–Teller (BET) theory was used to measure the surface areas of the clays. The powder samples were degassed in a vacuum at 200 degrees Celsius for one hour. The swell index of the clay was determined using the ASTM D5890-02 protocol [9], as described in a previous study [10]. The thermal stability of the basalt fiber-reinforced polymer (BFRP) composites was evaluated using thermogravimetric analysis (TGA). The TGA was performed using an SQT 600 model, heating the samples from 50 to 700 degrees Celsius at a rate of 10 degrees Celsius per minute under a nitrogen atmosphere. The fracture surfaces of the tensile specimens were examined using a high-resolution field emission scanning electron microscope (HR-FESEM, MERLIN (Carl Zeiss), Oberkochen, Germany). To prevent charging, the samples were coated with platinum.

### 2.5. Mechanical and Tribological Properties

Tensile tests were conducted at a rate of 0.2 mm per minute, following the ASTM D638-14 [11] standard. Friction and wear tests were carried out using a NEO-TRIBO MPW Friction and Wear Tester manufactured by Neoplus Inc. (Taejeon, Republic of Korea). The test specimen had a diameter of 40 mm. A load of 10 kg was applied using the ball-on-disk method at a temperature of 25 degrees Celsius. An SUS304 ball-type tip with a track radius of 11.5 mm was used in the test. The rotation speed was maintained at 30 revolutions per minute, and the total test time was 7200 s. The wear track depth was further analyzed using a surface profilometer (Dektak 150, Veeco, New York, NY, USA). The tests were performed for 100 s with a force of 1 milligram and a resolution of 0.200 μm per sample within the 6000-micrometer measurement range.

## 3. Results and Discussion

### 3.1. XRD and FTIR

The XRD spectra in Figure 1 contain peaks at 2θ = 19.79°, which represent the layered silicate structure of bentonite. The peak (001) at 2θ = 6.04° in the spectrum of the ARB clay corresponds to the interplanar (d) spacing of 14.50 Å. This interplanar space contains alkali cations. These cations are replaced with silane molecules, and, consequently, the interplanar spacing increases to 18.67 Å. The non-exfoliated silicate layers are represented by the peak at 2θ = 9.30°. Upon heating to 700 °C, bentonite loses its interlayer water, and its d spacing at (001) decreases to 9.68 Å. Moreover, calcination imparts a partially amorphous nature to the clays (increased noise in the XRD spectrum) [12]. Figure 2 shows the FTIR spectra of the ARB and modified bentonite clays. The peaks at 516.3 cm^−1^ and 922 cm^−1^ correspond to Al-O-Si bending and Al-Al-OH bending, respectively. The addition of silane groups on the surface of the bentonite clay is confirmed by the peak at 1560 cm^−1^ (NH_3_^+^ symmetric flexing) [13]. The peaks at 3432 and 3633 cm^−1^ are attributed to O-H stretching [14]. The peak at 1637 cm^−1^ is attributed to the O-H bending of the interlayer water [15]. The peaks corresponding to interlayer water disappear after the completion of the chemical and thermal modifications.

### 3.2. Specific Surface Area and Swell Index

The BET_surface area_ values of the bentonite clays are presented in Figure 3. The surface areas of the STB and CB clays increased by 314% and 176%, respectively, compared to that of the ARB clay. Studies have demonstrated that the SSA is determined by the microporosity resulting from the quasi-crystalline overlap region in the interlayer [16]. The presence of a silane molecule increased the interlayer spacing (001) of bentonite from 14.50 Å to 18.67 Å, which could have increased its surface area. Additionally, the increased surface area of the CB clay could be attributed to its partially amorphous nature, as determined by the XRD analysis. Furthermore, studies have demonstrated that the interlayer spacing of bentonite clay increases at high temperatures owing to the increased thermal motion of the interlayer species [17].

The swell indexes of the ARB, STB, and CB clays are depicted in Figure 4. The strong affinity of the ARB clay caused it to absorb a significant volume of water and hydrate the interlayer alkali cation, which led to a massive increase in the clay volume. The swell index of the ARB clay was 20 mL/2 g. The swell indexes of the modified clays were smaller than 10 mL/2 g, but they could not be quantified using the present setup. The data showed a significant reduction in swelling after silane treatment and calcination, owing to the loss of the interlayer alkali cation and the exfoliation of the aluminosilicate layers, followed by dehydroxylation (removal of interlayer water). In addition, the clear appearance of the CB and STB clays indicated their decreased hydrophilicity.

### 3.3. Thermogravimetric Analysis

The TGA data of the BF/bentonite/epoxy composites are presented in Figure 5. All composites underwent one-step degradation, albeit at different degradation onset temperatures. The degradation onset temperatures for the ARB, CB, and STB clay composites showed minimal variation, differing by less than three degrees (332.77 °C, 335.08 °C, and 333.28 °C, respectively). This indicates that the inclusion of the CB and STB clays provided only a slight improvement in the thermal stability of the composites. The high SSA and amorphous nature of the CB clay enhanced its physiochemical interactions with the epoxy matrix, while the presence of silane molecules in the interlayer of the STB clay acted as a heat barrier and provided thermal stability to the composite at high temperatures [7]. The weight losses of the CB and STB composites were 22.81% and 13.40% lower, respectively, than that of the ARB composite, owing to the enhanced interaction and bonding between the modified clays and BF–epoxy composites. The modified clay surfaces contained reactive functional groups that imparted thermal stability to the composites by consuming the free radicals generated during epoxy degradation, and the increased surface area acted as a shield that prevented the evaporation of the degraded components, thereby increasing the residence time of the composites at high temperatures.

### 3.4. Tensile Strength and Analysis of Fracture Surface

The ultimate tensile strength (UTS) values of the BF/bentonite/epoxy composites are illustrated in Figure 6. The UTS of the ARB clay-reinforced composite was 248 MPa. Meanwhile, the UTS values of the STB and CB clay-reinforced composites were 21% and 48% higher, respectively, representing significant increases. After the silane treatment and calcination, the interaction between the epoxy matrix, clay, and BF was strengthened owing to an increase in the surface area of the clay and a decrease in its water affinity. In addition, the STB and CB clays behaved as nano-clays owing to the exfoliation of the silicate layers, which improved their dispersion in the epoxy matrix and strengthened their interfacial interactions. The transmission electron microscopy (TEM) images in Figure 7 show increased transmission through the modified bentonite clays, further explaining the exfoliation of the silicate layers. The structure of bentonite is 2:1, and it consists of numerous stacked layers. After the silane treatment and calcination, this structure was de-stacked or exfoliated owing to the exchange of the interlayer cation with APTES and the collapse of the interlayer, respectively.

The increased interfacial interactions were further elucidated by performing an HR-FESEM analysis of the fracture surface, as illustrated in Figure 8. The number of interactions between the fibers, ARB clay, and the matrix decreased, as depicted in Figure 8A,B. The hydrophilic behavior and small surface area of the ARB clay caused agglomeration in the epoxy matrix and weak interfacial adhesion. In addition, the smooth surface of BF further confirmed the poor adhesion between the matrix and the fibers, which led to increased fiber pullout during fracture. By contrast, the fracture surface of the CB clay-reinforced composite in Figure 8C,D and that of the STB-clay-reinforced composite in Figure 8E contained fewer voids and exhibited reduced fiber pullout. In addition, strong adhesion was observed between the fiber surface and the epoxy matrix, which further explained the increased mechanical strength of these composites.

### 3.5. Tribological Properties

The tribological properties of the composites are illustrated in Figure 9. The coefficients of friction of the ARB, STB, and CB clay-reinforced composites were 0.523, 0.514, and 0.45, respectively. The STB clay-reinforced composite exhibited a slight improvement, whereas the coefficient of friction of the CB clay-reinforced composite decreased by 14% owing to increased interactions between the amorphous clay and epoxy matrix. The destruction of the crystalline structure of the clay at high temperatures caused the surface of the CB clay-reinforced composite to be smoother than those of the ARB and STB clay-reinforced composites. The exfoliated aluminosilicate layers slid and created lubrication between the steel ball and composite surface, which reduced the friction coefficient. A surface profilometer was used to further investigate the wear properties of the material. The cross-sectional wear track pattern, wear width, wear depth, and wear volume are depicted in Figure 9B–E, respectively. The wear volume and wear depth were calculated using the following equations [18,19]:V=2πrA
V=κ×F×S
where V, A, F, and S denote the wear volume, cross-sectional area, normal load, and sliding distance, respectively, and κ denotes the wear rate. The wear depth indicates the maximum penetration of the wear track, with the CB composite showing a significantly lower depth of 0.0214 mm compared to 0.151 mm for ARB and 0.05 mm for STB, suggesting superior wear resistance in the CB composite. Similarly, the wear width, which measures the lateral extent of the wear track, is narrower for the CB (2.421 mm) and STB (2.325 mm) composites than for ARB (3.161 mm), further indicating its better structural integrity under wear conditions among the modified composites. The wear volume, reflecting the total material loss due to wear, is dramatically reduced in the CB composite (1.9043 mm^3^) compared to ARB (20.128 mm^3^), with STB also demonstrating a significantly lower wear volume (3.12 mm^3^), showcasing the effectiveness of silane treatment and calcination in enhancing durability. Finally, the wear rate, which measures the material loss per unit load and distance, is the lowest for the CB composite at 0.00008428 mm^3^/Nm, indicating superior wear efficiency, while both the CB and STB composites exhibit markedly reduced wear rates compared to ARB (0.00087309 mm^3^/Nm for ARB and 0.0001353 mm^3^/Nm for STB). The composites with higher wear volumes exhibited higher wear rates and, therefore, higher wear loss. Overall, the results in Table 1 highlight the effectiveness of both modifications in significantly improving the wear performance of the bentonite-reinforced composites, with the CB composite.

The heat generated in the wear test increased the localized temperature of the composite’s contact surface, thereby weakening the polymer chains. The increased BET_surface area_ and reduced hydrophilicity of the modified clays improved their dispersion in the epoxy matrix, resulting in the distribution of the localized temperature over a larger area. The CB and STB clay-reinforced composites exhibited significantly improved wear properties owing to their improved compatibility with the epoxy matrix. Calcination imparted a partially amorphous nature to the bentonite clay, while silane modification significantly altered its surface area and swelling behavior. Hence, the compatibility and interfacial interactions increased with the addition of the CB and STB clays [20].

## 4. Conclusions

The modification of bentonite clay via silane treatment and calcination resulted in significant improvements in the mechanical and tribological properties of basalt fiber-reinforced epoxy (BFRP) composites, although the thermal stability gains were modest. Key findings include the following.

Thermal stability: The thermal degradation onset temperatures of the ARB-, CB-, and STB-reinforced composites increased only marginally, by less than 3 °C, with the CB and STB clays offering slight improvements in stability due to enhanced clay–epoxy interactions.Mechanical strength: The ultimate tensile strength (UTS) of the CB clay-reinforced composites increased by 48%, while STB composites showed a 21% increase compared to ARB composites. This was attributed to the better dispersion of the clays, increased surface area, and improved interfacial bonding between the clay, epoxy, and basalt fibers.Tribological properties: The wear volume decreased significantly, by 90% in CB composites and 84% in STB composites, due to the amorphous nature of calcined bentonite and the surface modification from the silane treatment. These changes reduced the hydrophilicity and enhanced the compatibility with the epoxy matrix, leading to better wear resistance.

In conclusion, while the inclusion of modified bentonite clays had a modest impact on the thermal stability, the significant improvements in the tensile strength and wear resistance underscore their potential to enhance the performance of multiscale fiber-reinforced composites in demanding applications.

## Figures and Tables

**Figure 1 materials-17-04760-f001:**
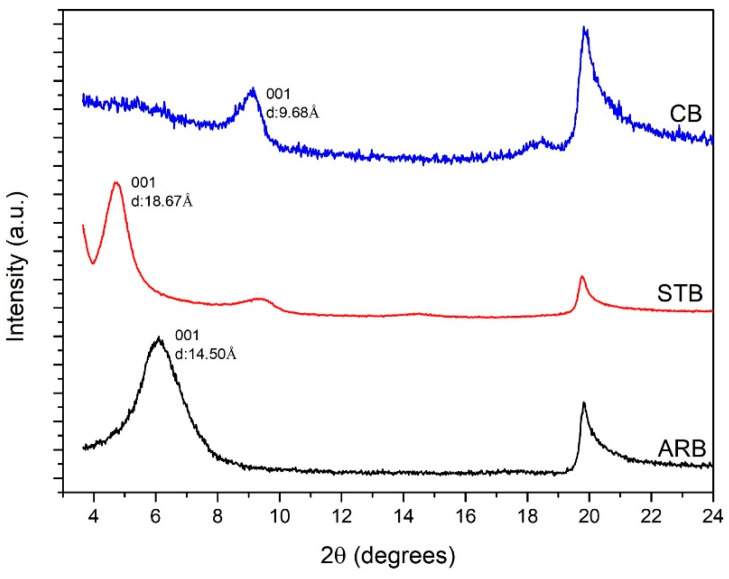
XRD spectra of the as-received bentonite (ARB), silane-modified bentonite (STB), and calcined bentonite (CB) clays.

**Figure 2 materials-17-04760-f002:**
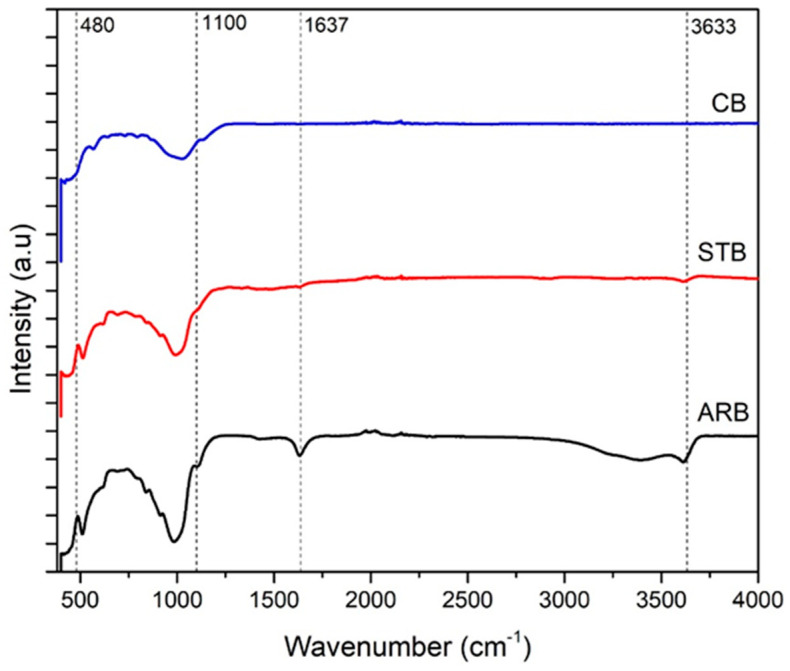
FTIR spectra.

**Figure 3 materials-17-04760-f003:**
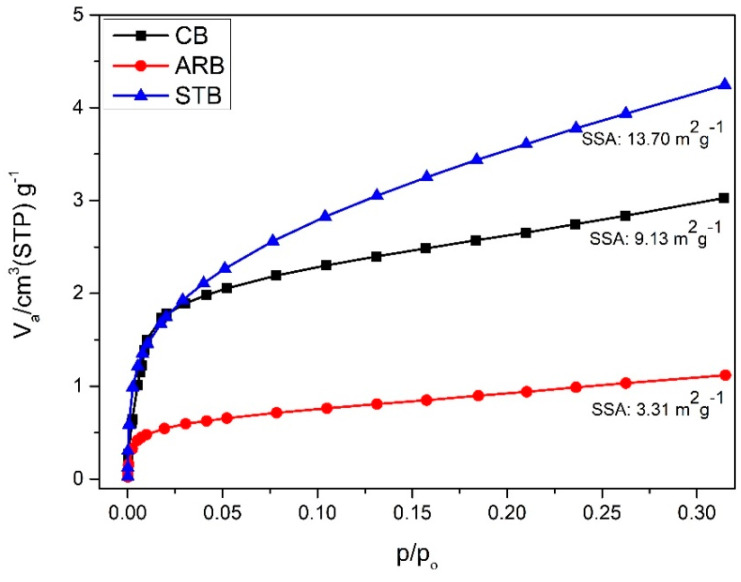
BET_surface area_ of clays.

**Figure 4 materials-17-04760-f004:**
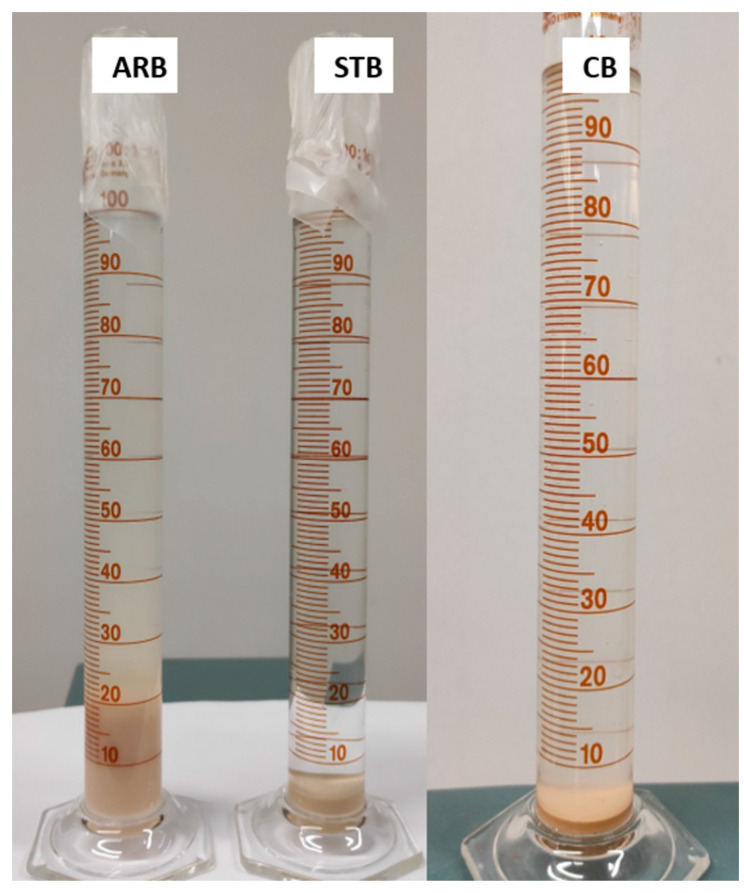
Swell index of bentonite clay.

**Figure 5 materials-17-04760-f005:**
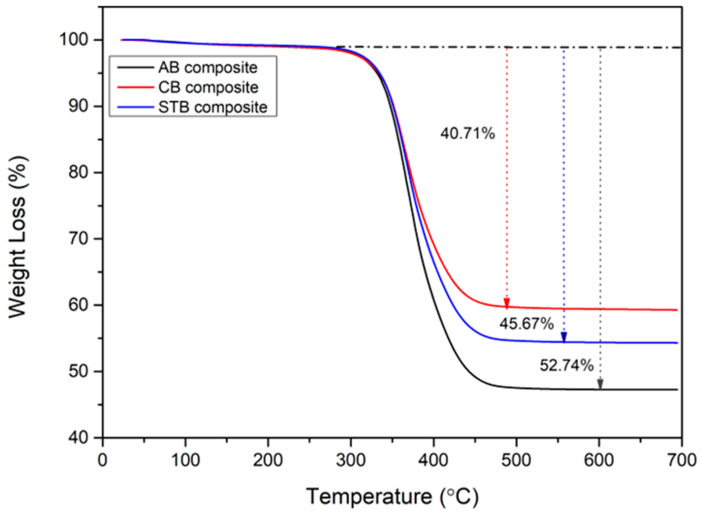
TGA of BF/bentonite/epoxy composites.

**Figure 6 materials-17-04760-f006:**
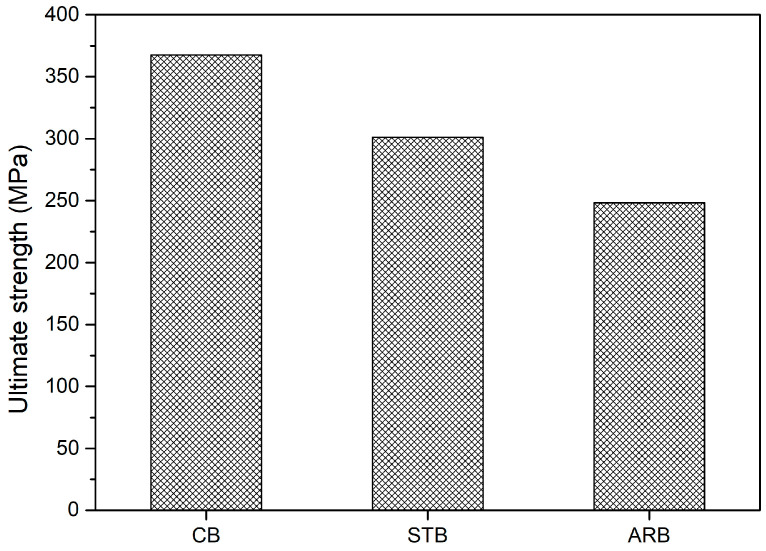
Tensile strength of BF/bentonite/epoxy composites.

**Figure 7 materials-17-04760-f007:**
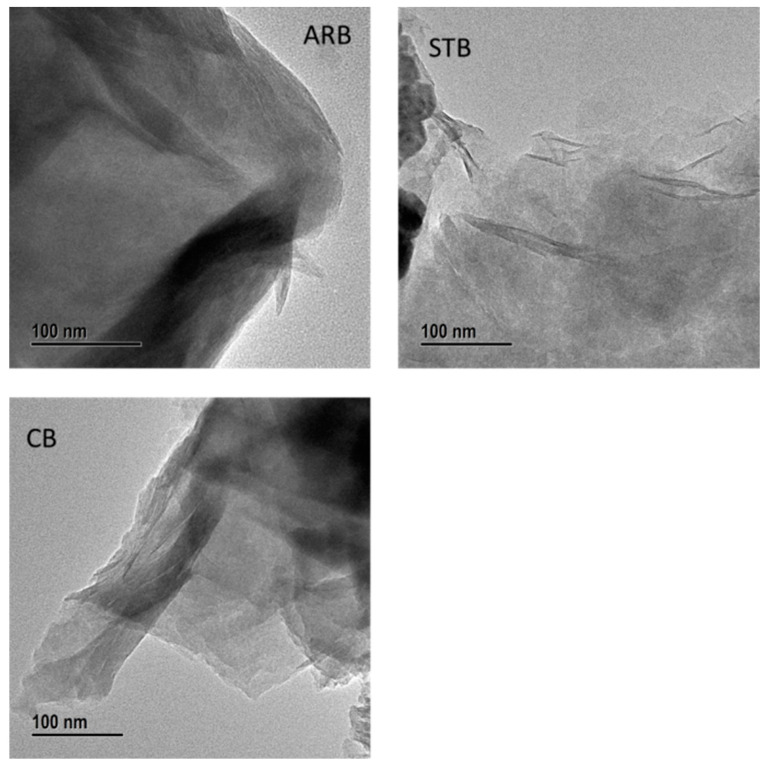
Transmission electron microscopy (TEM) images of the as-received and modified clays.

**Figure 8 materials-17-04760-f008:**
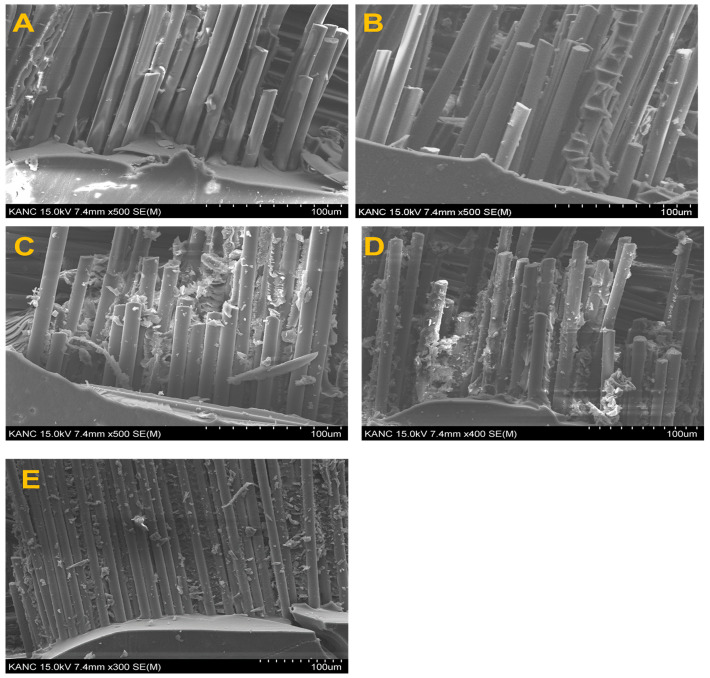
HR-FESEM of fracture surfaces of composites: (**A**,**B**) ARB, (**C**,**D**) CB, and (**E**) STB.

**Figure 9 materials-17-04760-f009:**
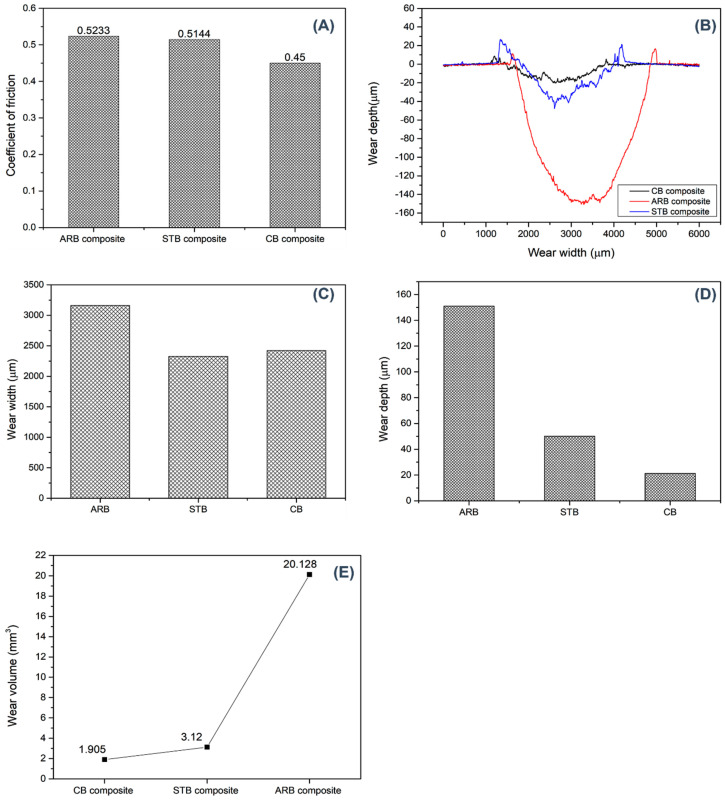
Tribological properties: (**A**) coefficient of friction, (**B**) wear pattern (cross-section), (**C**) wear width, (**D**) wear depth, and (**E**) wear volume.

**Table 1 materials-17-04760-t001:** Tribological parameters of the composites.

	CB Clay-Reinforced Composite	ARB Clay-Reinforced Composite	STB Clay-Reinforced Composite
Wear Depth (mm)	0.0214	0.151	0.05
Wear Width (mm)	2.421	3.161	2.325
Wear Volume (mm^3^)	1.9043	20.128	3.12
Wear Rate (mm^3^/Nm)	0.00008428	0.00087309	0.0001353

## Data Availability

Data is contained in the article.

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
