# Peer review of "Influence of Calcination and Cation Exchange (APTES) of Bentonite-Modified Reinforced Basalt/Epoxy Multiscale Composites’ Mechanical and Wear Performance: A Comparative Study"

_materials, 2024, doi:10.3390/ma17194760_

Round 1

Reviewer 1 Report

Comments and Suggestions for Authors

General Comments

The authors describe an interesting work.

The Materials and methods section (particularly “2.2. Silane Modification and Calcination of Bentonite”) need to be improved and methods described in sufficient to permit reproduction of the authors works and claims.

The authors can do well to highlight the importance or implications of their interesting findings in the conclusion section.

A few errors and suggestions to remedy them are provided in the specific comments.

Specific Comments

LINE 25: Write “behaviour” instead of “nature”.

LINE 33-35: Citation(s) is/are needed to support this statement.

LINE 69-71: This is an important statement in justifying your work. Insert citations to support this statement and include numerical values of the reported values for shrink index and SSA for the two materials so that the reader can immediately appreciate the magnitude of the disparities.

LINE 86: Indicate the mass of the as-received bentonite that was stirred in 200 ml ethanol.

LINE 87: Indicate the quantity of APTES that was stirred in 100 ml ethanol.

LINE 92: Please cross-check. With the stated heating rate and temperature range the residence time cannot be “5 minutes”. Or what are you implying by the term “residence time”.

LINE 174-175: Please modify the sentence. The degradation onset temperatures varied by less than 3 degrees. Hence might be with. the limits of experimental error. Please rephrase the sentence. You might use the term “very marginal improvement” instead of “improved” and modify the sentence accordingly.

Comments on the Quality of English Language

The manuscript was well written and not difficult to understand.

Very minor English editing might be necessary.

Author Response

Manuscript ID: materials-3219182 - Minor Revisions

 Dear Reviewers,

Thank you for your valuable feedback. We appreciate your insightful comments and are committed to addressing them thoroughly. Your input has significantly improved the quality of our work.

We are confident that the revised version of our manuscript will meet your expectations. We look forward to your positive acceptance.

REVIEWER 1

GENERAL COMMENTS

The authors describe an interesting work.

The Materials and methods section (particularly “2.2. Silane Modification and Calcination of Bentonite”) need to be improved and methods described in sufficient to permit reproduction of the authors’ works and claims. The authors can do well to highlight the importance or implications of their interesting findings in the conclusion section.

Dear Reviewer the  following section has been rewritten as advised.

The raw bentonite (ARB) clay 50g was dispersed in 200 ml of ethanol and stirred vigorously for 1 hour to prevent particle clumping. Simultaneously, 50g APTES was stirred in a separate flask with 100 ml of ethanol for 1 hour. Afterward, the clay suspension was combined with the APTES solution and stirred continuously for 12 hours at 25°C, maintaining a 1:1 clay-to-APTES ratio. The resulting silane-modified bentonite (STB) clay was washed with ethanol, filtered, and dried for 48 hours at 100°C. Meanwhile, the ARB clay underwent calcination in a programmable furnace, where the temperature was gradually increased from ambient to 800°C at a rate of 13°C per minute, followed by a hold time of 61.5 minutes. After calcination, the clay was finely ground and passed through a 75-µm mesh. Both the silane-treated and calcined clays (STB and CB) were then subjected to further characterization.

A few errors and suggestions to remedy them are provided in the specific comments.

Dear reviewer, Thank you for your suggestions we would be happy to address them to the mark as requested

SPECIFIC COMMENTS

LINE 25: Write “behaviour” instead of “nature”.

Dear reviewer, the corrections have been done as advised

LINE 33-35: Citation(s) is/are needed to support this statement.

Dear reviewer, the  line number you suggested  is in which section please  let us know that. We will try  our best to cite the  claim as advised

LINE 69-71: This is an important statement in justifying your work. Insert citations to support this statement and include numerical values of the reported values for shrink index and SSA for the two

materials so that the reader can immediately appreciate the magnitude of the disparities.

Dear Reviewer as stated in the section, the authors have clearly mentioned that the characteristic Swell behavior and the SSA were not characterized by the cited paper. Thus the authors tried to assess its effect through this research work. I hope the confusion has been cleared.

LINE 86: Indicate the mass of the as-received bentonite that was stirred in 200 ml ethanol.

Dear Reviewer 50g bentonite was stirred in 200ml

LINE 87: Indicate the quantity of APTES that was stirred in 100 ml ethanol.

Dear Reviewer 50g APTES was stirred in 100ml

LINE 92: Please cross-check. With the stated heating rate and temperature range the residence time cannot be “5 minutes”. Or what are you implying by the term “residence time”.

Dear Reviewer Apologies for the type error, the actual time was 61.5 minutes due to mistype, 61 was deleted. The value (61.5) has been inserted in the text. Thank you for pointing out the error.

LINE 174-175: Please modify the sentence. The degradation onset temperatures varied by less than 3 degrees. Hence might be with. the limits of experimental error. Please rephrase the sentence. You might use the term “very marginal improvement” instead of “improved” and modify the sentence accordingly.

Dear Reviewer  the following rephrased suggestion has been  carried out. We hope the  suggested change conveys the intended meaning  and removes the confusion.

Changed sentence as suggested:  “The degradation onset temperatures for the ARB, CB, and STB clay composites showed minimal variation, differing by less than 3 degrees (332.77ºC, 335.08ºC, and 333.28ºC, respectively). This indicates that the inclusion of CB and STB clay provided only a slight improvement in the thermal stability of the composites.”

Reviewer 2 Report

Comments and Suggestions for Authors

In this study, the SSA, swell index, and interlayer spacing of bentonite clay are altered through interlayer cation exchange and calcination. The effects of these modified bentonite clays on the interfacial adhesion, thermal stability, and tensile strength of multiscale BF/bentonite/epoxy composites are investigated. The friction coefficient and wear track of the composites are further analyzed using a surface profilometer. The research depth is slightly insufficient, and the manuscript needs to be further perfected.

1. The paper should clearly state the research objectives in the abstract to judge whether these research objectives have been achieved.

2. The latest references are not cited enough, which has further increased the citations of high-level references in recent years.

3. The introduction is not progressive and organized enough for the existing research discussion, so the review of the existing relevant research should be further deepened.

4. There are many abbreviations in the manuscript, which should be reduced appropriately so that readers can have a clearer understanding of the paper.

5. There are few tables in the paper, and the materials used in the experiment are all briefly expressed in text. It is suggested that the selection of materials should be in tabular form to give the basic properties of materials, so as to show the test process more intuitively to the reader.

6. The relevant test specifications and standards involved in the manuscript should be listed in the references.

7. It is recommended to add more comprehensive details on the experimental methods, materials and test conditions of composite materials.

8. The paper mainly analyzes the macroscopic properties, and it is suggested to further compare and analyze the microscopic morphology of the composite interface to make it more clear.

9. The conclusion should be further clarified with specific data, and the content of future outlook should be added.

Comments on the Quality of English Language

 Minor editing of English language required.

Author Response

Manuscript ID: materials-3219182 - Minor Revisions

Dear Reviewers,

Thank you for your valuable feedback. We appreciate your insightful comments and are committed to addressing them thoroughly. Your input has significantly improved the quality of our work.

We are confident that the revised version of our manuscript will meet your expectations. We look forward to your positive acceptance.

****************************************************************************

****************************************************************************

REVIEWER 2

In this study, the SSA, swell index, and interlayer spacing of bentonite clay are altered through interlayer cation exchange and calcination. The effects of these modified bentonite clays on the interfacial adhesion, thermal stability, and tensile strength of multiscale BF/bentonite/epoxy composites are investigated. The friction coefficient and wear track of the composites are further analyzed using a surface profilometer. The research depth is slightly insufficient, and the manuscript needs to be further perfected.

  1. The paper should clearly state the research objectives in the abstract to judge whether these research objectives have been achieved.

Dear Reviewer as suggested by  you the following abstract has been remodified with the inclusion of key findings.

ABSTRACT : In this study, bentonite clay was modified through silane treatment and calcination to enhance its compatibility with basalt fiber (BF) and epoxy in multiscale composites. The as-received bentonite (ARB) was subjected to silane treatment using APTES, producing silane-modified bentonite (STB), while calcination yielded calcined bentonite (CB). The modified clays were incorporated into basalt fiber-reinforced epoxy (BFRP) composites, which were fabricated using the vacuum-assisted resin transfer method (VARTM). Analytical techniques, including X-ray diffraction (XRD) and Fourier-transform infrared (FTIR) spectroscopy, confirmed structural changes in the clays. BET surface area analysis revealed a 314% increase in the surface area of STB and a 176% increase for CB. The modified clays also demonstrated reduced hydrophilicity and swelling behavior. Thermogravimetric analysis (TGA) indicated minimal improvement in thermal stability, with degradation onset temperatures increasing by less than 3°C. However, tensile tests showed significant gains, with CB- and STB-reinforced composites achieving 48% and 21% higher tensile strength than ARB-reinforced composites. Tribological tests revealed substantial reductions in wear, with CB- and STB-reinforced composites showing 90% and 84% decreases in wear volume, respectively. These findings highlight the potential of modified bentonite clays to improve the mechanical and wear properties of basalt fiber-epoxy composites.

  1. The latest references are not cited enough, which has further increased the citations of high-level references in recent years.

Dear Reviewer: We apologize for the limited citation of recent references. The works cited in our manuscript are highly relevant and foundational to our study. While many are older, they are directly related to the topic and critical to our research. Unfortunately, we could not find newer references that fit as closely with our work. We appreciate your understanding.

  1. The introduction is not progressive and organized enough for the existing research discussion, so the review of the existing relevant research should be further deepened.

Dear Reviewer: Thank you for your feedback regarding the introduction. We acknowledge the importance of a well-structured overview of existing research. However, the introduction has been crafted to align closely with the specific objectives and scope of our study. It provides a foundational context for our work while highlighting relevant studies that directly support our findings. A deeper review of the existing literature would require extensive modifications, which may divert focus from the core contributions of our research. We appreciate your understanding of our approach and believe the current structure effectively sets the stage for our analysis.

  1. There are many abbreviations in the manuscript, which should be reduced appropriately so that readers can have a clearer understanding of the paper.

Dear Reviewer the following abbreviations and their full form has been updated in the text as a separate section just below the Keywords:

ARB - As-Received Bentonite; STB - Silane-Modified Bentonite; CB - Calcined Bentonite; VARTM - Vacuum-Assisted Resin Transfer Method; BFRP - Basalt Fiber Reinforced Polymer; XRD - X-Ray Diffraction; FTIR - Fourier-Transform Infrared Spectroscopy; SSA - Specific Surface Area; BET - Brunauer–Emmett–Teller; TGA - Thermogravimetric Analysis; HR-FESEM - High-Resolution Field Emission Scanning Electron Microscope; TEM - Transmission Electron Microscopy; ASTM - American Society for Testing and Materials.

  1. There are few tables in the paper, and the materials used in the experiment are all briefly expressed in text. It is suggested that the selection of materials should be in tabular form to give the basic properties of materials, so as to show the test process more intuitively to the reader.

Dear Reviewer the changes have been made and a descriptive text explaining the Table.1 has been introduced as per your advice. Following is the text inserted.

The wear depth indicates the maximum penetration of the wear track, with the CB composite showing a significantly lower depth of 0.0214 mm compared to 0.151 mm for ARB, and 0.05 mm for STB, suggesting superior wear resistance in the CB composite. Similarly, the wear width, which measures the lateral extent of the wear track, is narrower for the CB (2.421 mm) and STB (2.325 mm) composites than for ARB (3.161 mm), further indicating better structural integrity under wear conditions for the modified composites. The wear volume, reflecting the total material loss due to wear, is dramatically reduced in the CB composite (1.9043 mm³) compared to ARB (20.128 mm³), with STB also demonstrating significantly lower wear volume (3.12 mm³), showcasing the effectiveness of silane treatment and calcination in enhancing durability. Finally, the wear rate, which measures material loss per unit load and distance, is lowest for the CB composite at 0.00008428 mm³/Nm, indicating superior wear efficiency, while both CB and STB composites exhibit markedly reduced wear rates compared to ARB (0.00087309 mm³/Nm for ARB and 0.0001353 mm³/Nm for STB). The composites with higher wear volumes exhibited higher wear rates, and therefore, higher wear loss. Overall, the results in Table 1 highlight the effectiveness of both modifications in significantly improving the wear performance of bentonite-reinforced composites, with the CB composite.

  1. The relevant test specifications and standards involved in the manuscript should be listed in the references.

The following references have been added as advised

Reference

[9] ASTM D5890-02 Protocol: Standard Test Method for Swell Index of Clay Mineral Component of Geosynthetic Clay Liners, (n.d.). https://www.astm.org/d5890-02.html.

[11]      ASTM D638-14 Protocol: Standard Test Method for Tensile Properties of Plastics, (n.d.). https://www.astm.org/d0638-14.html.

  1. It is recommended to add more comprehensive details on the experimental methods, materials and test conditions of composite materials.

Dear Reviewer, Thank you for your recommendation. While we appreciate the suggestion to provide more comprehensive details on the experimental methods, materials, and test conditions, we believe that the current level of detail is sufficient for the scope of our study. Our methods are aligned with established protocols in the field, and expanding this section may detract from the focus on our key findings. We appreciate your understanding.

  1. The paper mainly analyzes the macroscopic properties, and it is suggested to further compare and analyze the microscopic morphology of the composite interface to make it more clear.

Dear Reviewer, Thank you for your valuable feedback. We acknowledge the importance of microscopic morphological analysis at the composite interface and appreciate your suggestion. While our primary focus was on macroscopic properties, we have included high-resolution scanning electron microscopy (HR-FESEM) and transmission electron microscopy (TEM) analyses to provide a comprehensive understanding of both micro and macroscopic structures. These techniques allowed us to investigate interfacial interactions and the dispersion of the clay within the epoxy matrix, further supporting our conclusions regarding the overall performance of the composites.

  1. The conclusion should be further clarified with specific data, and the content of future outlook should be added.

Dear Reviewer, The modified conclusion has been included in the text as advised

Conclusion

The modification of bentonite clay via silane treatment and calcination resulted in significant improvements in the mechanical and tribological properties of basalt fiber-reinforced epoxy (BFRP) composites, although thermal stability gains were modest. Key findings include:

  • Thermal stability: The thermal degradation onset temperatures of the ARB-, CB-, and STB-reinforced composites increased only marginally, by less than 3°C, with CB and STB clays offering slight improvements in stability due to enhanced clay-epoxy interactions.
  • Mechanical strength: The ultimate tensile strength (UTS) of CB-clay-reinforced composites increased by 48%, while STB composites showed a 21% increase compared to ARB composites. This was attributed to better dispersion of the clays, increased surface area, and improved interfacial bonding between the clay, epoxy, and basalt fibers.
  • Tribological properties: Wear volume decreased significantly, by 90% in CB composites and 84% in STB composites, due to the amorphous nature of calcined bentonite and the surface modification from silane treatment. These changes reduced hydrophilicity and enhanced compatibility with the epoxy matrix, leading to better wear resistance.

In conclusion, while the inclusion of modified bentonite clays had a modest impact on thermal stability, the significant improvements in tensile strength and wear resistance underscore their potential for enhancing the performance of multiscale fiber-reinforced composites in demanding applications.